# Photocatalytic Reduction of CO$_2$ to Methanol by Cu$_2$O/TiO$_2$ Heterojunctions

S.-P. Cheng, L.-W. Wei and H.-Paul Wang *

Department of Environmental Engineering, National Cheng Kung University, Tainan 70101, Taiwan;
pingc2102@gmail.com (S.-P.C.); wei771210@yahoo.com.tw (L.-W.W.)
* Correspondence: wanghp@ncku.edu.tw; Tel.: +886-6-275-7575 (ext. 65832)

**Abstract:** The conversion of CO$_2$ to low-carbon fuels using solar energy is considered an economically attractive and environmentally friendly route. The development of novel catalysts and the use of solar energy via photocatalysis are key to achieving the goal of chemically reducing CO$_2$ under mild conditions. TiO$_2$ is not very effective for the photocatalytic reduction of CO$_2$ to low-carbon chemicals such as methanol (CH$_3$OH). Thus, in this work, novel Cu$_2$O/TiO$_2$ heterojunctions that can effectively separate photogenerated electrons and holes were prepared for photocatalytic CO$_2$-to-CH$_3$OH. More visible light-active Cu$_2$O in the Cu$_2$O/TiO$_2$ heterojunctions favors the formation of methanol under visible light irradiation. On the other hand, under UV-Vis irradiation for 6 h, the CH$_3$OH yielded from the photocatalytic CO$_2$-to-CH$_3$OH by the Cu$_2$O/TiO$_2$ heterojunctions is 21.0–70.6 μmol/g-catalyst. In contrast, the yield of CH$_3$OH decreases with an increase in the Cu$_2$O fraction in the Cu$_2$O/TiO$_2$ heterojunctions. It seems that excess Cu$_2$O in Cu$_2$O/TiO$_2$ heterojunctions may lead to less UV light exposure for the photocatalysts, and may decrease the conversion efficiency of CO$_2$ to CH$_3$OH.

**Keywords:** photocatalysis; CO$_2$; Cu$_2$O; TiO$_2$; p-n heterojunctions; methanol





## 1. Introduction

In recent years, the significant rise of greenhouse gas CO$_2$ concentrations on the earth causing serious problems has received much attention. There are major challenges in recycling high-thermal stability CO$_2$, which may involve severe reaction conditions (high pressures or high temperatures) with extra energy consumption that may lead to the additional formation of CO$_2$. Thus, the recycling of CO$_2$ into low-carbon chemicals or fuels using solar energy is considered an economically attractive and environmentally friendly route. The desired photocatalysts for the photocatalytic reduction of CO$_2$ to chemicals or fuels under mild conditions are being developed to achieve the goal of CO$_2$ recycling [1–3].

Titanium dioxide (TiO$_2$), an n-type semiconductor, has been used for the photocatalytic reduction of CO$_2$ to chemicals such as formic acid, formaldehyde, methane and methanol [4,5]. TiO$_2$ has shown great potential for various photocatalytic reactions, mainly due to its chemical stability, nontoxicity, high oxidation efficiency and environmentally friendly nature [6]. However, because of its fairly wide bandgap (3.2 and 3.0 eV for anatase and rutile phases, respectively), TiO$_2$ can only be activated by ultraviolet (UV) light, equivalent to about 5% of natural solar light. A variety of strategies, such as metal ion doping, cation or anion doping, and coupling with narrow-bandgap semiconductors, have been developed to extend absorption into the visible light region [7–10]. The doping of anions (e.g., N, F, S, and C) onto TiO$_2$ could shift the absorption edge to a relatively low energy, and its photo-response into the visible spectrum [11–14]. Cation-, anion- or metal ion-doped TiO$_2$ could lead to better solar energy harvesting in the visible light region; however, this still suffers from relatively high photogenerated electron and hole recombination rates, causing difficulties in engineering applications [15–17].

CO$_2$ may be activated by a one-electron transfer step and form a ·CO$_2^-$ radical ion. The ·CO$_2^-$ may be reduced to yield a hydroxyformyl radical (·COOH), which recombines

a hydrogen radical ($H^+$) and an electron ($e^-$) to form formic acid [18]. In the following step, formic acid accepts $H^+$ and $e^-$ to form formaldehyde. Formic acid and formaldehyde seem to be formed prior to methanol generation. Thus, the key points that control the photocatalytic $CO_2$-to-$CH_3OH$ reaction may include reaction conditions, photocatalyst activity, bandgap energy, light source and process parameters. To effectively suppress the rapid recombination of photoexcited electrons and holes, a heterojunction structure could facilitate electron migration [19,20]. Cuprous oxide ($Cu_2O$), a typical p-type semiconductor, has wide application prospects in solar cells, photocatalysis, and hydrogen evolution reactions (HER) [21]. $Cu_2O$, with a bandgap energy of 2.0–2.2 eV, could effectively harvest visible light for photocatalysis. However, while the photocatalytic $CO_2$-to-$CH_3OH$ reaction facilitated by $Cu_2O$ is thermodynamically feasible, its $CH_3OH$ yield suffers from the low solar conversion efficiency [22]. By the heterojunction between the p-type $Cu_2O$ and n-type $TiO_2$, the recombination of photo-excited charges could be effectively retarded and facilitate photocatalytic reactions [23,24]. In this work, novel $Cu_2O/TiO_2$ heterojunctions were thus prepared by a simple soft chemical method as the visible light photocatalysts used for the enhanced photocatalytic reduction of $CO_2$ to methanol.

## 2. Materials and Methods

$Cu_2O$ was prepared by the facile soft chemical method (Figure 1). Briefly, $CuCl_2$ (97%, Merck, Kenilworth, NJ, USA) (10.1 mmol) was dispersed in a NaCl solution (5 M) (100 mL) with a dispersant (polyethylene glycol 20,000 (Sigma-Aldrich, Burlington, MA, USA) (0.025 mmol)), which was stirred at 298 K for 1 h. $Na_3PO_4$ (96%, Sigma-Aldrich, USA) (9.76 mmol) was added to the solution and stirred for 1 h. The $Cu_2O$ was centrifuged and cleaned with distilled water and ethanol three times. Titanium butoxide ($Ti(OBu)_4$) (97%, Sigma-Aldrich, USA) and $Cu_2O$ at the $X_{Cu2O}$ mole fractions ($Cu_2O/(Cu_2O + TiO_2)$) of 0.1, 0.2 and 0.5 were mixed in deionized water, and were then centrifuged, dried at 378 K for 4 h, and heated at 723 K under $N_2$ flow (99.99%) (20 mL/min) for 2 h to obtain the $Cu_2O/TiO_2$ heterojunctions used for photocatalysis experiments.

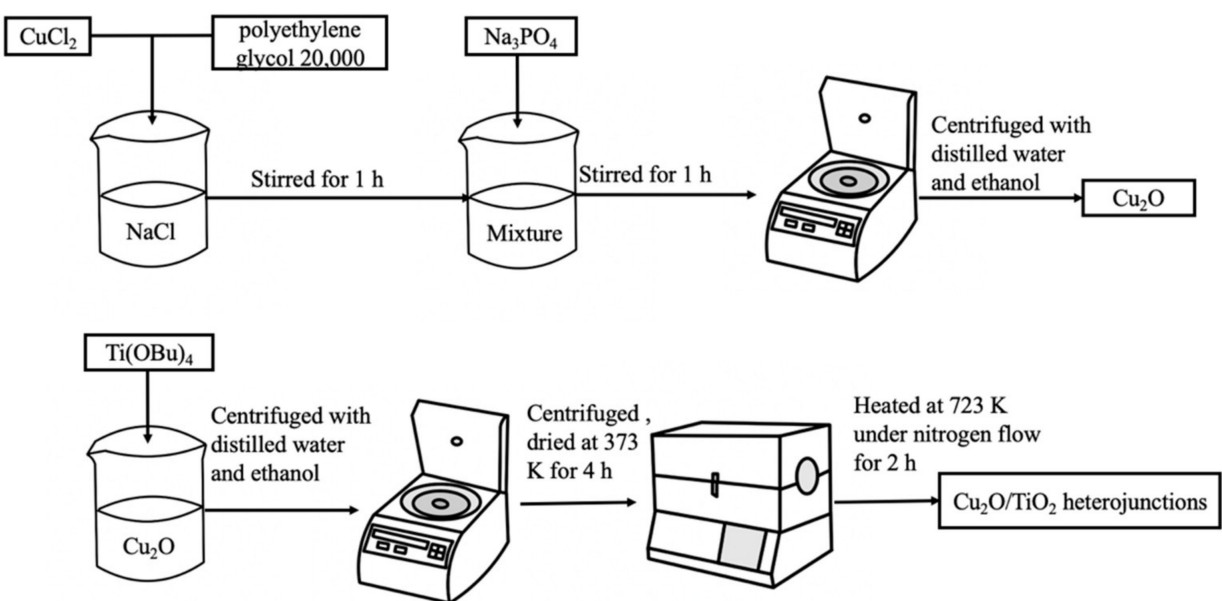

**Figure 1.** Preparation procedure for the $Cu_2O/TiO_2$ heterojunction photocatalysts.

The crystalline structures of the $Cu_2O$, $TiO_2$ and $Cu_2O/TiO_2$ heterojunctions were determined by X-ray diffraction (D8, Discover with Gadds, Bruker AXS Gmbh). The crystalline sizes of the $Cu_2O$, $TiO_2$ and $Cu_2O/TiO_2$ heterojunction photocatlysts were calculated by the Scherrer equation ($t = k\lambda/B\cos\theta$) using the Jade software. The images of the $Cu_2O/TiO_2$ heterojunctions were investigated by scanning electron microscopy (SEM)

equipped with an energy-dispersive X-ray spectrometer (EDS) (AURIGA) and scanning transmission electron microscopy (JEOL JEM-2100F CS STEM). The room temperature photoluminescence spectra of the photocatalysts were determined on the LabRAM HR (Horiba Jobin Yvon, Palaiseau, France) using the 325 nm excitation wavelength. The diffuse reflection absorption spectra of the photocatalysts at 200–800 nm were studied on a UV–visible spectrophotometer (Varian, Cary 100, Palo Alto, CA, USA). $BaSO_4$ was used as the standard in the absorption spectroscopic experiments. The bandgap energy was studied via the Kubelka–Munk equation ($\alpha h\nu = A(h\nu\text{-}Eg)^n$). The specific surface area, pore size and pore volume distribution of the photocatalysts were measured on a nitrogen adsorption–desorption analyzer (Micromeritics, ASAP 2020) using the Brunauer–Emmett–Teller (BET) and Barret–Joyner–Halenda (BJH) model. The zeta potential data were determined from the specific surface areas and active sites of the photocatalysts.

The photocatalytic experiments were carried out in a closed cylindrical quartz reactor to prevent oxygen/air access. The photocatalyst (0.1 g) was dispersed in a sodium hydroxide (0.025 M) aqueous solution (100 mL). Before the photocatalysis experiments, high-purity $CO_2$ was bubbled through the solution until the pH reached 7.00 at 298 K. A 300 W Xenon arc lamp (Burgeon Instrument Co., Ltd., Taoyuan City, Taiwan) with the light cut off ($\lambda > 400$ nm) by a filter (FSQ-CG400, Newport, Taipei, Taiwan) was used for the experiments on the photocatalytic reduction of $CO_2$ to methanol. The concentrations of the photocatalytic product methanol were measured by GC-MS (JEOL JMS-700 and Shimadzu, QP2010).

## 3. Results and Discussion

The XRD patterns of the photocatalysts are shown in Figure 2. The diffraction peaks at 29.6°, 36.5°, 42.4°, 61.4°, 73.6° and 77.5° correspond to the (110), (111), (200), (220), (311) and (222) phases of the crystalline $Cu_2O$ (JCPDS card No. 78-2076), respectively [25]. A high-intensity diffraction peak at 36.4° confirms the existence of $Cu_2O$ in the $Cu_2O/TiO_2$ heterojunctions. Other diffraction peaks at 25.3°, 37.8°, 48.0°, 53.9°, 55.0°, 62.7°, 68.8°, 70.3° and 75.0° can be indexed to the (101), (004), (200), (105), (211), (204), (116), (220) and (215) planes of $TiO_2$ (JCPDS card No. 71-1167), associated with the anatase phase [26,27], indicating that the $Cu_2O/TiO_2$ heterojunctions consist of anatase predominantly. Note that a peak at 38.7° related to the CuO(111) plane is also observed, suggesting the existence of a small amount of CuO. The crystalline sizes of the $TiO_2$, $Cu_2O$, and $Cu_2O/TiO_2$ heterojunctions derived by Scherrer's equation are 50–100, 50–70, and 40–100 nm, respectively.

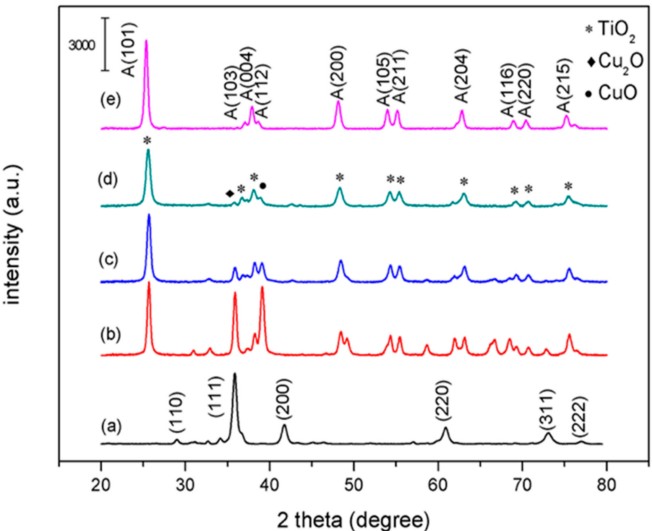

**Figure 2.** X-ray diffraction patterns of the (**a**) $Cu_2O$ and $Cu_2O/TiO_2$ heterojunctions with the $X_{Cu2O}$ fractions of (**b**) 0.5, (**c**) 0.2, (**d**) 0.1, and (**e**) $TiO_2$ nanoparticles.

The TEM images of the $TiO_2$ and $Cu_2O/TiO_2$ heterojunctions are shown in Figure 3. It is clear that the $Cu_2O$ (in the $Cu_2O/TiO_2$ heterojunction) and $TiO_2$ have nanoparticle diameters of ~5 and 70–130 nm, respectively. The presence of Cu, Ti, and O in the $Cu_2O/TiO_2$ heterojunction could be revealed by energy-dispersive X-ray (EDX) spectroscopy (see Figure 3c). The HRTEM image of the sample in Figure 3d shows lattice fringes spacing of 0.212 and 0.237 nm, corresponding to the (200) and (111) planes of $Cu_2O$, respectively [28]. The $TiO_2$ with high crystallinity has the d-spacings of 0.352 and 0.246 nm, related to the (100) and (004) planes of anatase $TiO_2$ [28].

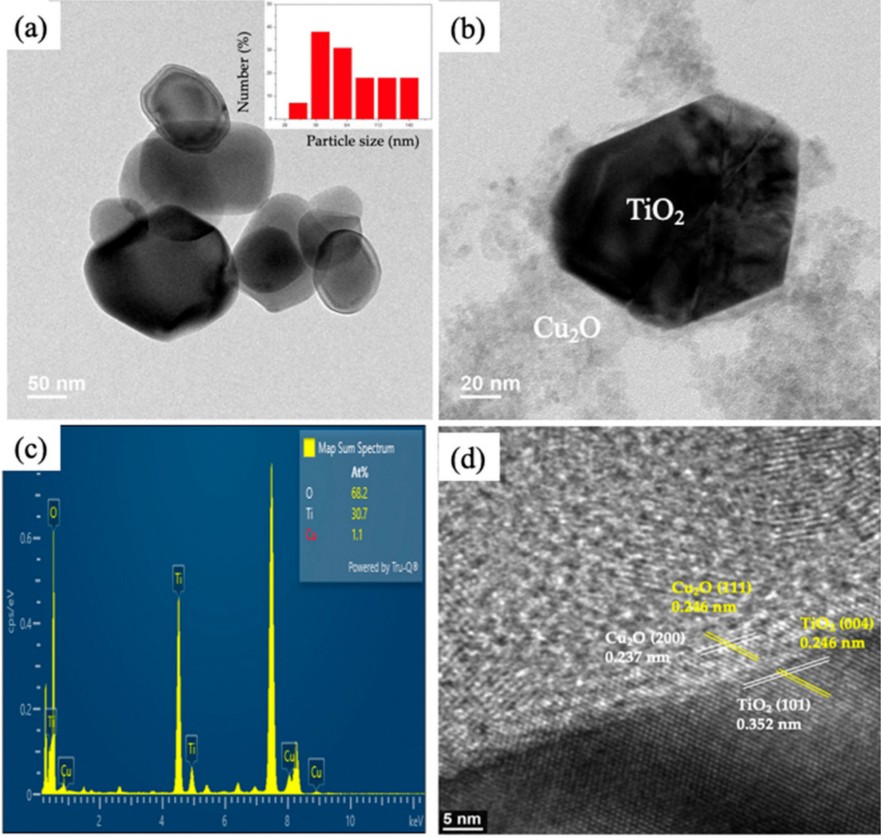

**Figure 3.** The TEM images of the (**a**) pristine $TiO_2$ and (**b**) $Cu_2O/TiO_2$ heterojunctions ($X_{Cu2O}$ = 0.5) with (**c**) EDS mapping, and (**d**) the HRTEM image.

The nitrogen adsorption–desorption isothermals of the pristine $TiO_2$ and $Cu_2O/TiO_2$ heterojunctions are shown in Figure 4A. The absorption isothermals of the $Cu_2O/TiO_2$ heterojunctions can be classified as type IV with H1 hysteresis loops, suggesting that they have a mesoporous structure. In Figure 4B, the $Cu_2O/TiO_2$ heterojunctions have greater pore volumes, with pore diameters between 10 and 40 nm, than the pristine $TiO_2$, possessing relatively high pore diameters of 30–70 nm. It seems that the smaller $Cu_2O$ nanoparticles may, to some extent, be incorporated into the pores of the $TiO_2$, which creates more internal surfaces in the interfaces of the $Cu_2O$ and $TiO_2$ nanoparticles. In Table 1, as expected, the $Cu_2O/TiO_2$ heterojunctions have relatively high specific surface areas (94–120 m$^2$/g) and small average pore diameters, which may benefit the photocatalytic reduction of $CO_2$ to $CH_3OH$.

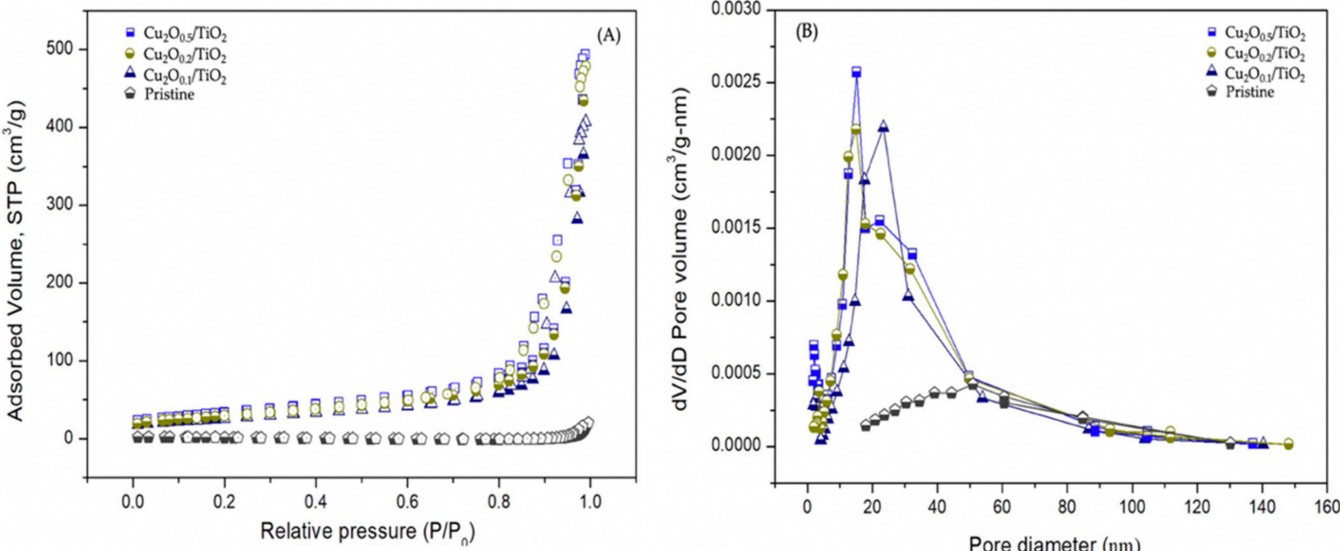

**Figure 4.** (**A**) $N_2$ absorption–desorption isothermals and (**B**) pore size distributions of the $Cu_2O/TiO_2$ heterojunctions with the $X_{Cu2O}$ fractions of 0.1, 0.2, 0.5, and pristine $TiO_2$ nanoparticles.

**Table 1.** The BET surface areas, average pore diameters, and zeta potentials of the pristine $TiO_2$ and $Cu_2O/TiO_2$ heterojunctions with the $X_{Cu2O}$ fractions of 0.1–0.5.

| Photocatalysts | $X_{Cu2O}$ Fractions | BET Surface Area ($m^2$/g) | BJH Average Pore Diameter (nm) | Zeta Potential (mV) |
|---|---|---|---|---|
| Pristine $TiO_2$ | 0 | 48 | 50 | 52.7 |
| | 0.1 | 94 | 23 | −21.1 |
| $Cu_2O/TiO_2$ | 0.2 | 105 | 15 | −26.6 |
| | 0.5 | 120 | 15 | −39.1 |

According to the linear regression analysis, the relationship between the zeta potentials (see in Table 1) and BET surfaces of the $Cu_2O/TiO_2$ heterojunctions and $TiO_2$ nanoparticles was well fitted ($R^2 > 0.9$). The zeta potential data were determined by the specific surface areas and active sites of the photocatalysts. The $Cu_2O/TiO_2$ heterojunctions with negative potential can provide more active sites for $CO_2$ reduction, suggesting that the $Cu_2O/TiO_2$ heterojunctions are feasible for photocatalytic $CO_2$-to-$CH_3OH$ reactions.

The diffuse reflectance ultraviolet–visible spectra of the photocatalysts were also determined. In Figure 5, the absorbance of the $Cu_2O/TiO_2$ heterojunctions in the range of 200–800 nm can be observed. Compared with $TiO_2$, the fundamental absorbance cuts at 400 nm, and the $TiO_2$ mixed with $Cu_2O$ reveals a significantly enhanced absorption in the visible light region. It is clear that $TiO_2$ with $Cu_2O$ causes a red-shift to 400–800 nm in the visible light range, possibly due to the forming of the $Cu_2O/TiO_2$ heterojunctions [28].

In Figure 6, the bandgap energies of the photocatalysts were determined by the Kubelka–Munk transforms [29]. The direct bandgaps of the $TiO_2$, $Cu_2O$ and $Cu_2O/TiO_2$ heterojunctions with the $X_{Cu2O}$ fractions of 0.1, 0.2 and 0.5 were estimated to be 3.20, 2.09, 3.03, 3.0 and 2.94 eV, respectively. It seems that the coupling of $Cu_2O$ with $TiO_2$ can effectively decrease the bandgap energies of both. The $Cu_2O/TiO_2$ heterojunctions turn out to be more photoactive than $TiO_2$ under visible light irradiation.



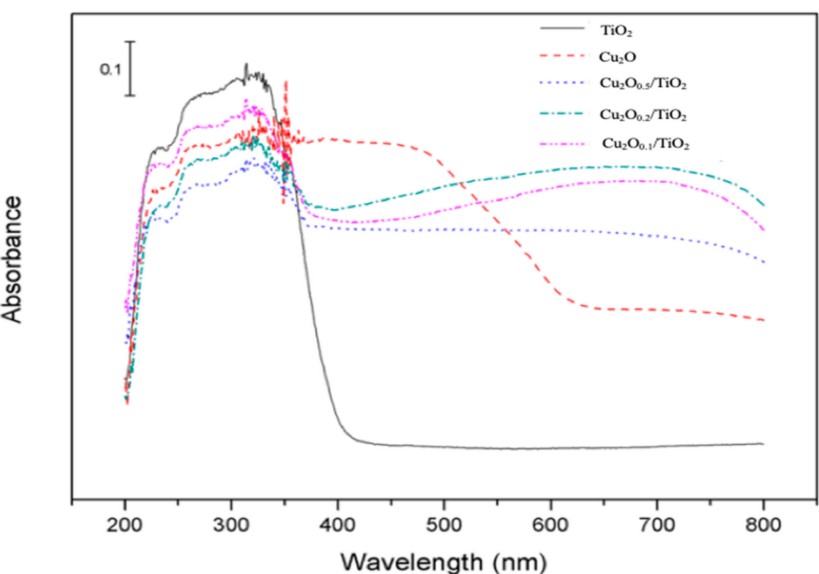

**Figure 5.** UV-Vis DR spectra of the $Cu_2O/TiO_2$ heterojunctions with the $X_{Cu2O}$ fractions of 0.1–0.5.

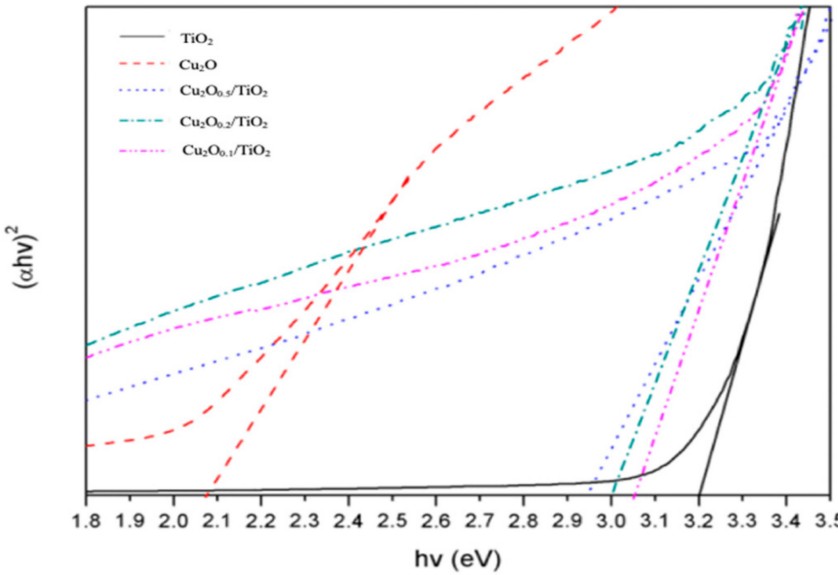

**Figure 6.** The Tauc plot of the $Cu_2O/TiO_2$ heterojunctions with the $X_{Cu2O}$ fractions of 0.1–0.5.

The charge separation efficiency of photoinduced electrons and holes is also one of the important factors in photocatalysis. The photoluminescence spectra can provide information on charge carrier trapping, migration and transfer [30]. The photoluminescence spectra of the $Cu_2O/TiO_2$ heterojunctions are shown in Figure 7. The photoluminescence intensity of the $Cu_2O/TiO_2$ heterojunctions is less than that of $TiO_2$. A clear quenching of the photoluminescence emission of the $Cu_2O/TiO_2$ heterojunctions is observed, especially for the $Cu_2O/TiO_2$ heterojunctions with the $X_{Cu2O}$ fraction of 0.2, which showed maximum quenching. Such quenching of the photoluminescence suggests that the separation of photogenerated electron and hole pairs in the $Cu_2O/TiO_2$ heterojunctions has been effectively improved. $Cu_2O$ can transfer the photogenerated holes from $TiO_2$ to inhibit the recombination of photogenerated electrons and holes significantly, and this may consequently lead to enhanced photocatalysis.

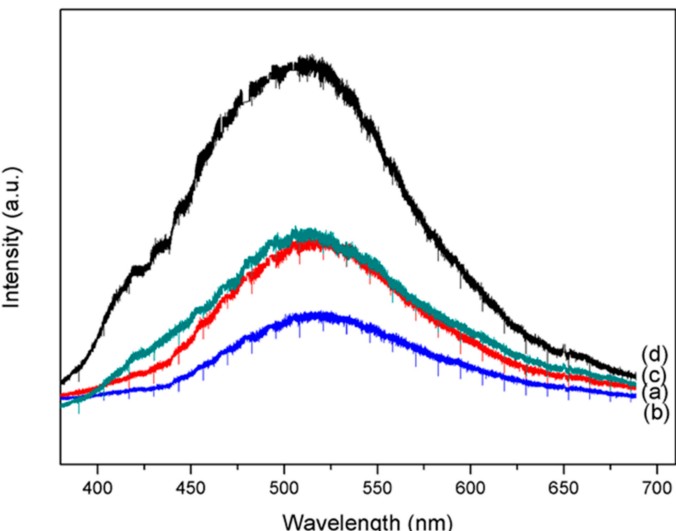

**Figure 7.** The photoluminescence spectra of the $Cu_2O/TiO_2$ heterojunctions with the $X_{Cu2O}$ fractions of of (a) 0.1, (b) 0.2 and (c) 0.5, and (d) pristine $TiO_2$ nanoparticles.

Figure 8A shows the yield of $CH_3OH$ from the photocatalytic reduction of $CO_2$ by the $Cu_2O/TiO_2$ heterojunctions under visible light irradiation. After 6 h of visible light irradiation, the yields of $CH_3OH$ photocatalyzed by the $Cu_2O/TiO_2$ heterojunctions with the $X_{Cu2O}$ fractions of 0.1, 0.2, and 0.5 are 9.29, 11.44, and 13.06 µmol/g-catalyst, respectively. Note that, as expected, $TiO_2$ is not very effective for the photocatalytic $CO_2$-to-$CH_3OH$ reaction. Additionally, more visible light-active $Cu_2O$ in the $Cu_2O/TiO_2$ heterojunctions favors the formation of methanol. On the other hand, under UV-Vis irradiation for 6 h, the $CH_3OH$ yielded from the photocatalytic $CO_2$-to-$CH_3OH$ reaction by the $Cu_2O/TiO_2$ heterojunctions is 21.0–70.6 µmol/g-catalyst (see Figure 8B). It is clear that the yields of $CH_3OH$ under UV-Vis irradiation are greater than those under visible irradiation. In contrast, the yield of $CH_3OH$ decreases with an increase in the $Cu_2O$ fraction in the $Cu_2O/TiO_2$ heterojunctions. It seems that excess $Cu_2O$ in the $Cu_2O/TiO_2$ heterojunctions may lead to less UV light exposure for the photocatalysts, and may reduce the conversion efficiency of $CO_2$ to $CH_3OH$.

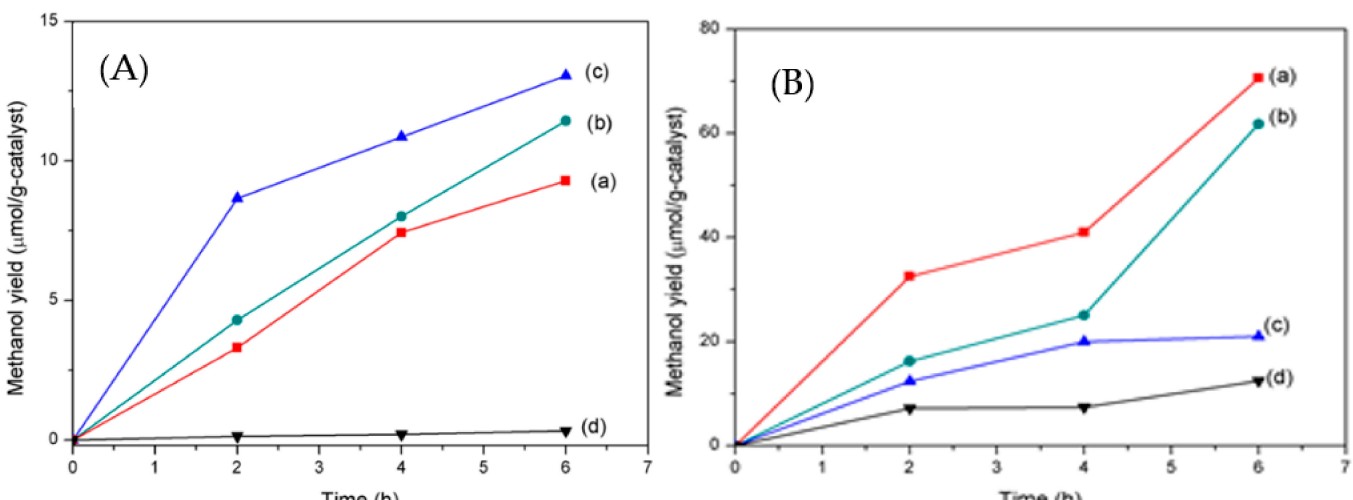

**Figure 8.** Photocatalytic reduction of $CO_2$ to methanol under (**A**) visible and (**B**) UV–visible irradiation by the $Cu_2O/TiO_2$ heterojunctions with the $X_{Cu2O}$ fractions of (a) 0.1, (b) 0.2, (c) and 0.5, and (d) $TiO_2$ nanoparticles.

The schematic diagram of the charge separation in the $Cu_2O/TiO_2$ heterojunction structure is depicted in Scheme 1. When the $Cu_2O/TiO_2$ heterojunctions are irradiated by visible light, only the electrons of $Cu_2O$ can be excited to the conduction band, and then move to the $TiO_2$, leading to the better separation of electron and hole pairs. However, photoexcited electrons in $Cu_2O$ and $TiO_2$ are excited to the conduction band when irradiated under UV–visible light, whereas the holes of $TiO_2$ may quickly transfer to the $Cu_2O$, which may reduce the recombination of photogenerated holes and electrons and promote the photocatalytic activity.

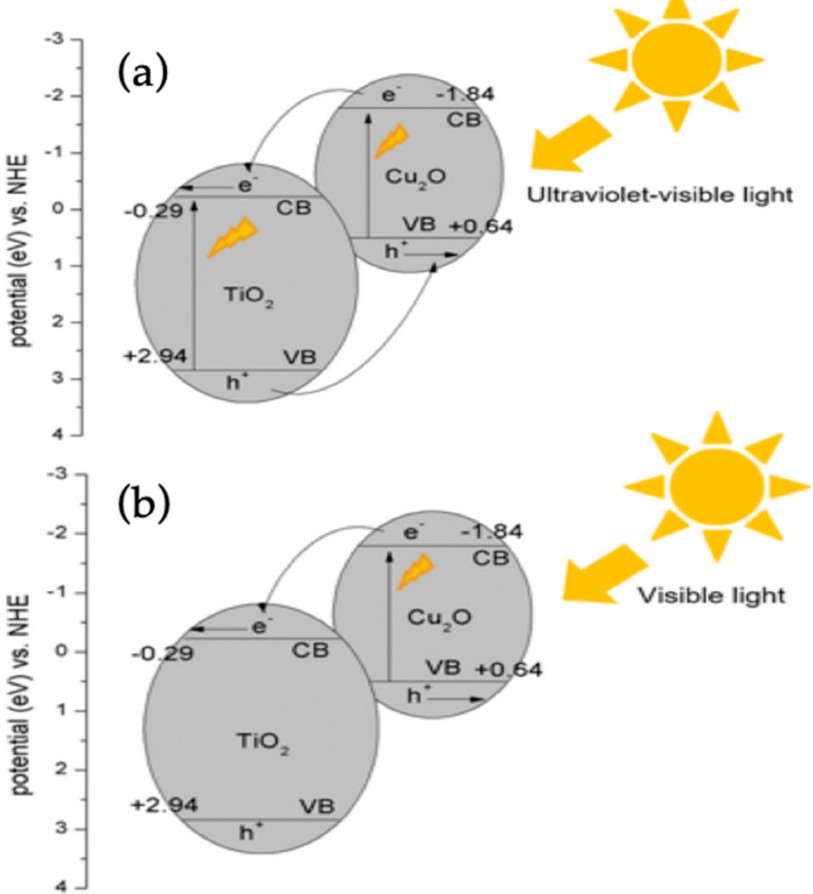

**Scheme 1.** The charge separation with the $Cu_2O/TiO_2$ heterojunctions under (**a**) ultraviolet–visible and (**b**) visible light irradiation.

$CO_2$ can be activated by a one-electron transfer step to form·$CO_2^-$, which may be reduced to yield the hydroxyformyl radical (COOH), which recombines with a hydrogen radical ($H^+$) and an electron ($e^-$) to form formic acid and formaldehyde (which seem to be formed prior to the $CH_3OH$ generation) [18]. Thus, the key points that control the photocatalytic $CO_2$-to-$CH_3OH$ reaction may include reaction conditions, photocatalyst activity, bandgap energy, light source and process parameters, and the comparison between different methods is shown in Table 2 [31–38]. It is clear that the $Cu_2O/TiO_2$ heterojunctions prepared in this work offer much better $CH_3OH$ yields under visible light irradiation.

**Table 2.** Method comparison for photocatalytic $CO_2$-to-$CH_3OH$ reaction.

| Photocatalyst | Light Source | Bandgap Energy (eV) | Reactions | $CH_3OH$ Yield (μmol/g/h) | Ref. |
|---|---|---|---|---|---|
| $Cu_2O/TiO_2$ | UV<br>Vis | 2.9–3.0 | 100 mg photocatalysts, $CO_2$ in deionized water | 9–13<br>12–70 | This work |
| $Co/TiO_2$ | UV | - | 100 mg photocatalysts, $CO_2$ in $NaHCO_3$ (1 M) | 0.05 | [31] |
| Anatase $TiO_2$ | Vis | 2.9–3.2 | 500 mg photocatalysts, $CO_2$ in deionized water | 2.74 | [32] |
| $SnO_2/g$-$C_3N_4$ | UV | - | 20 mg photocatalysts, $CO_2$ in water vapor | 0.02 | [33] |
| $rGO/ZnO$ | Vis | 2.8 | 100 mg photocatalysts, $CO_2$ in water vapor | 0.42 | [34] |
| $rGO/Cu_2O$ | UV | 2.7–2.8 | 100 mg photocatalysts, $CO_2$ in NaOH (1 M) | 8.77 | [35] |
| $CQDs/Cu_2O$ | Vis | 2.4–2.6 | 150 mg photocatalysts, $CO_2$ in deionized water | 1.96 | [36] |
| $ZnTe/SrTiO_3$ | UV | 2.8–3.4 | 20 mg photocatalysts, $CO_2$ in deionized water | 0.75 | [37] |
| $ZnO/g$-$C_3N_4$ | UV | 2.6–3.0 | 10 mg photocatalysts, $CO_2$ in deionized water | 0.06 | [38] |

## 4. Conclusions

The novel $Cu_2O/TiO_2$ heterojunction photocatalysts prepared by a simple soft chemical method have relatively high specific surface areas and small average pore diameters, which may benefit the photocatalytic reduction of $CO_2$ to $CH_3OH$. The $Cu_2O$ in conjunction with $TiO_2$ decreases its bandgap energy, and extends the absorption to the visible light region. The p-n-type heterojunction can effectively suppress charge carrier recombination. After the 6 h photocatalytic reduction, 9–13 and 21–76 μmol/g-catalyst of methanol can be yielded under visible and UV-Vis irradiation, respectively. The comparison between different methods suggests that the $Cu_2O/TiO_2$ heterojunctions prepared in this work offer much better $CH_3OH$ yields under visible and UV light irradiation.

**Author Contributions:** S.-P.C. designed the concept and drafted the manuscript. L.-W.W. provided support of the literature search and manuscript revision. H.-P.W. supervised the research work and revised the manuscript. All authors have read and agreed to the published version of the manuscript.

**Funding:** This research was funded by the Taiwan Ministry Science and Technology and EPA.

**Institutional Review Board Statement:** Not applicable.

**Informed Consent Statement:** Not applicable.

**Data Availability Statement:** The data presented in this study are available on request from the corresponding author.

**Acknowledgments:** The financial supports of the Taiwan Ministry Science and Technology and EPA for sponsoring this project (MoST 108-2221-E-006-165-MY3, MoST 109-2221-E-006 -042 -MY3, MoST 110-2221-E-006-107-MY2, and EPA 110GA00008001047) are gratefully acknowledged.

**Conflicts of Interest:** The authors declare no conflict of interest.

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
