# Peer review of "Photocatalytic Reduction of CO2 to Methanol by Cu2O/TiO2 Heterojunctions"

_sustainability, doi:10.3390/su14010374_

Round 1

Reviewer 1 Report

This paper examines the synthesis and characterization of Cu-TiO2 heterojunctions and their usage for the photocatalytic reduction of carbon dioxide to methanol using visible or uv/vis irradiation. The subject belongs to sustainability however, the work needs some revisions

Some specific points:

  • Analytical part: more information is needed regarding photocatalytic experiments. For example, I suppose that the quartz reactor was closed/covered, preventing oxygen /air. Since the authors demonstrate the difference between the visible and the simulated solar light, they must first refer to how they perform the experiment with only visible light (using 400 nm cut off filter?) and then perform actinometry for UV-VIS and vis irradiation. The nominal power has no meaning
  • Although the physicochemical characterization is good, what is missing is the BET and the zpc of the synthesized catalyst, which may affect the discussion about efficiency. In general, the authors ignore the “physical” part of the process and relate the efficiency only to the different separation or light absorbance
  • I’m sure that plenty of papers have been published regarding the photocatalytic reduction of CO2 to methanol. Can the authors prepare a table to compare the yield for published works and the work presented here?

Author Response

Review of Manuscript ID: Sustainability- 1474103

Title: Photocatalytic reduction of CO2 to methanol by Cu2O/TiO2 hetero-junctions

Responses to the Comments of Reviewers

The manuscript was carefully re-examined and revised to responses and fulfilled the reviewers’ comments. Additional data, better presentation of Figures, and more detailed discussion are included in the revised manuscript. All the authors have seen the revised manuscript with the responses to reviewers’ comments and approved submission. The manuscript was presented at the International Conference on Innovations in Energy Engineering & Cleaner Production (IEECP’21) and caught notable attention. Thus, we believe this manuscript will be of interest to the Sustainability audience.

Reviewer #1

Comment #1: Analytical part: more information is needed regarding photocatalytic experiments. For example, I suppose that the quartz reactor was closed/covered, preventing oxygen /air. Since the authors demonstrate the difference between the visible and the simulated solar light, they must first refer to how they perform the experiment with only visible light (using 400 nm cut off filter?) and then perform actinometry for UV-VIS and vis irradiation. The nominal power has no meaning. Although the physicochemical characterization is good, what is missing is the BET and the zpc of the synthesized catalyst, which may affect the discussion about efficiency. In general, the authors ignore the “physical” part of the process and relate the efficiency only to the different separation or light absorbance. I’m sure that plenty of papers have been published regarding the photocatalytic reduction of CO2 to methanol. Can the authors prepare a table to compare the yield for published works and the work presented here?

Responses to Comment #1:

The photocatalysis experiments were carried out in a closed cylindrical quartz reactor to prevent oxygen/air.

A 300 W Xenon arc lamp (Burgeon Instrument Co., Ltd) having the light cutoff (λ >400 nm) by a filter (FSQ-CG400, Newport) was used for the experiments of photocatalytic reduction of CO2 to methanol.

As suggested, additional physicochemical characterization (BET and zpc) of the photocatalysts was studied. The nitrogen adsorption-desorption isothermals of the pristine TiO2 and Cu2O/TiO2 heterojunctions are shown in Figure 4(A). The absorption isothermals of the Cu2O/TiO2 heterojunctions can be classified as type IV with H1 hysteresis loops, suggesting that they have mesoporous structure. In Figure 4(B), the Cu2O/TiO2 heterojunctions have greater pore volumes with the pore diameters between 10-40 nm than the pristine TiO2 possessing relatively high pore diameters of 30-70 nm. It seems that the smaller Cu2O nanoparticles may, to some extent, be incorporated in the pores of the TiO2, which creates more internal surfaces in the interfaces of Cu2O and TiO2 nanoparticles. In Table 1, as expected, the Cu2O/TiO2 heterojunctions have relatively high specific surface areas (94-120 m2/g) and small average pore diameter, which may benefit the photocatalytic reduction of CO2 to methanol.

By the linear regression analysis, the relationship between zeta potential and BET surfaces of the Cu2O/TiO2 heterojunctions and TiO2 nanoparticles were well fitted (R2 >0.9) (see in Table 1). The zeta potential data were determined by the specific surface areas and active sites of the photocatalysts. The Cu2O/TiO2 having negative potential can provide more active sites for CO2 reduction, suggesting that the Cu2O/TiO2 heterojunctions are feasible for photocatalytic CO2-to-methanol reactions.

CO2 can be activated by a one-electron transfer step to form ·CO2- that may be reduced to yield the hydroxyformyl radical (·COOH), which recombines with a hydrogen radical (H+) and an electron (e-) to form formic acid [18]. Formic acid can accept H+ and e- to form formaldehyde. Formic acid and formaldehyde seem to be formed prior to the methanol generation. Thus, the key points that control the photocatalytic CO2-to-CH3OH may include reaction conditions, photocatalyst activity, bandgap energy, light source and process parameters, and the comparison between different methods is shown in Table 2 [31-38]. It is clear that the Cu2O/TiO2 heterojunctions prepared in this work have better CH3OH yields under visible light irradiation.

 (The related statements are also described in the revised manuscript)

Figure. 4 (A) N2 absorption-desorption isothermals and (B) pore size distributions of the Cu2O/TiO2 heterojunctions having the XCu2O fractions of (a) 0.1, (b) 0.2, (c) 0.5 and (d) pristine TiO2.

Table 1. The BET surface areas, average pore diameters, and zeta potential of the pristine TiO2 and Cu2O/TiO2 heterojunctions having the XCu2O fractions of 0.1-0.5.

Photocatalysts

XCu2O

fractions

BET surface

Area (m2/g)

BJH average pore

diameter (nm)

Zeta potential (mV)

Pristine TiO2

0

48

50

52.7

Cu2O/TiO2

0.1

94

23

-21.1

Cu2O/TiO2

0.2

105

15

-26.6

Cu2O/TiO2

0.5

120

15

-39.1

Table 2. Method comparison for photocatalytic CO2-to-CH3OH.

Photocatalyst

Light source

Bandgap

Energy (eV)

Reactions

CH3OH yield

(µmol/g/h)

Ref.

Cu2O/TiO2

UV

Vis

2.9-3.0

100 mg photocatalysts,

CO2 in deionized water

9-13

12-70

This work

Co/TiO2

UV

-

100 mg photocatalysts,

CO2 in NaHCO3 (1 M)

0.05

[31]

Anatase TiO2

Vis

2.9-3.2

500 mg photocatalysts,

CO2 in deionized water

2.74

[32]

SnO2/g-C3N4

UV

-

20 mg photocatalysts,

CO2 in water vapor

0.02

[33]

rGO/ZnO

Vis

2.8

100 mg photocatalysts,

CO2 in water vapor

0.42

[34]

rGO/Cu2O

UV

2.7-2.8

100 mg photocatalysts,

CO2 in NaOH (1 M)

8.77

[35]

CQDs/Cu2O

Vis

2.4-2.6

150 mg photocatalysts,

CO2 in deionized water

1.96

[36]

ZnTe/SrTiO3

UV

2.8-3.4

20 mg photocatalysts,

CO2 in deionized water

0.75

[37]

ZnO/g-C3N4

UV

2.6-3.0

10 mg photocatalysts,

CO2 in deionized water

0.06

[38]

Author Response

Review of Manuscript ID: Sustainability- 1474103

Title: Photocatalytic reduction of CO2 to methanol by Cu2O/TiO2 hetero-junctions

Responses to the Comments of Reviewers

The manuscript was carefully re-examined and revised to responses and fulfilled the reviewers’ comments. Additional data, better presentation of Figures, and more detailed discussion are included in the revised manuscript. All the authors have seen the revised manuscript with the responses to reviewers’ comments and approved submission. The manuscript was presented at the International Conference on Innovations in Energy Engineering & Cleaner Production (IEECP’21) and caught notable attention. Thus, we believe this manuscript will be of interest to the Sustainability audience.

Reviewer #2

Comment #1: L55: “Chemical structure of the Cu2O,....” – Crystalline Structure

 L72-78: Refer the JCPDS card# in XRD discussion and cite the relevant papers.

Improve the scale-bar visibility in Fig 2 (a-d). 2(d)- text not legible.

L123: what type of TiO2 does the author refer here in PL study (L124)?

Response:

    L55 (L73 (revised)): ”Crystalline structure of the Cu2O, TiO2 and Cu2O/TiO2 heterojunctions were determined by X-ray Diffraction (D8, Discover with Gadds, Bruker AXS Gmbh).”

    L72-78 (L98-106 (revised)): ” The diffraction peaks at 29.6°, 36.5°, 42.4°, 61.4°, 73.6° and 77.5° correspond to (110), (111), (200), (220), (311) and (222) phases of the crystalline Cu2O (JCPDS card No. 78-2076), respectively [25]. A high intensity diffraction peak at 36.4° confirms the existence of Cu2O in the Cu2O/TiO2 heterojunctions. Other diffraction peaks at 25.3°, 37.8°, 48.0°, 53.9°, 55.0°, 62.7°, 68.8°, 70.3° and 75.0° can be indexed to (101), (004), (200), (105), (211), (204), (116), (220) and (215) planes of TiO2 (JCPDS card No. 71-1167) associated with the anatase phase [26], indicating that Cu2O/TiO2 heterojunctions consist of anatase predominantly.”

        The scale-bar visibility in Figure 3(a-d) was improved. The HRTEM image of the sample in Figure 2(d) shows lattice fringes spacing of 0.212 and 0.237 nm, corresponding to the (200) and (111) plane of Cu2O, respectively [28]. The TiO2 with a highly crystallinity has the d-spacings of 0.352 and 0.246 nm related to the (100) and (004) planes of anatase TiO2 [28].

L123 (L168 (revised)): “Figure. 5 The photoluminescence spectra of the Cu2O/TiO2 heterojunctions having the XCu2O fractions of (a) 0.1, (b) 0.2, (c) 0.5, and (d) pristine TiO2 nanoparticles.”

(The related statements are also described in the revised manuscript)

Figure. 3 The TEM images of the (a) pristine TiO2 and (b) Cu2O/TiO2 heterojunctions (XCu2O=0.5) with (c) EDS mapping and (d) HRTEM image.

Comment #2: Authors should clearly explain the performance of their synthesized Cu2O-TiO2 heterojunctions w.r.to published results. The novelty of this work has not been explained clearly. Authors need to cover literature and provide citations of relevant studies throughout the manuscript.

Response:

To effectively suppress the rapid recombination of photoexcited electron and hole, the heterojunction structure could facilitate the electron migration [19, 20]. Cuprous oxide (Cu2O), a typical p-type semiconductor, has wide applications in solar cells, photocatalysis, and Hydrogen evolution reaction (HER) [21]. Cu2O having bandgap energy of 2.0-2.2 eV could effectively harvest visible light for photocatalysis. Although, the photocatalytic CO2-to-CH3OH by Cu2O is thermodynamically feasible, its CH3OH yield suffered from the low solar-conversion efficiency [22]. By heterojunction between p-type Cu2O and n-type TiO2, the recombination of photo-excited charges could be effectively retarded and facilitated photocatalytic reactions [23, 24]. In this work, the novel Cu2O/TiO2 heterojunctions were thus prepared by a simple soft chemical method as the visible light photocatalysts for enhanced photocatalytic reduction of CO2 to methanol.

CO2 can be activated by a one-electron transfer step to form ·CO2- that may be reduced to yield the hydroxyformyl radical (·COOH), which recombines with a hydrogen radical (H+) and an electron (e-) to form formic acid [18]. Formic acid can accept H+ and e- to form formaldehyde. Formic acid and formaldehyde seem to be formed prior to the methanol generation. Thus, the key points that control the photocatalytic CO2-to-CH3OH may include reaction conditions, photocatalyst activity, bandgap energy, light source and process parameters, and the comparison between different methods is shown in Table 2 [31-38]. It is clear that the Cu2O/TiO2 heterojunctions prepared in this work have better CH3OH yields under visible light irradiation.

(The related statements are also described in the revised manuscript)

Comment #3: Introduction is short. Authors should elaborate the hypothesis and objectives of this work. Also, more background literature study should be incorporated. For examples., the following recent articles on hetero junction photocatalytic system should be referred in this work.

ChemCatChem 2018, 10, 3305

Applied Surface Science, Volume 520, 1 August 2020, 146344

Journal of Hazardous Materials 168(1):484-92, 2009

Response:

The Introduction section was extended to elaborate the hypothesis and objectives of this work, and more background literature studies was incorporated: “To effectively suppress the rapid recombination of photoexcited electron and hole, the heterojunction structure could facilitate the electron migration [19, 20]. Cuprous oxide (Cu2O), a typical p-type semiconductor, has wide applications in solar cells, photocatalysis, and Hydrogen evolution reaction (HER) [21]. Cu2O having bandgap energy of 2.0-2.2 eV could effectively harvest visible light for photocatalysis. Although, the photocatalytic CO2-to-CH3OH by Cu2O is thermodynamically feasible, its CH3OH yield suffered from the low solar-conversion efficiency [22]. By heterojunction between p-type Cu2O and n-type TiO2, the recombination of photo-excited charges could be effectively retarded and facilitated photocatalytic reactions [23, 24]. In this work, the novel Cu2O/TiO2 heterojunctions were thus prepared by a simple soft chemical method as the visible light photocatalysts for enhanced photocatalytic reduction of CO2 to methanol.

CO2 can be activated by a one-electron transfer step to form ·CO2- that may be reduced to yield the hydroxyformyl radical (·COOH), which recombines with a hydrogen radical (H+) and an electron (e-) to form formic acid [18]. Formic acid can accept H+ and e- to form formaldehyde. Formic acid and formaldehyde seem to be formed prior to the methanol generation. Thus, the key points that control the photocatalytic CO2-to-CH3OH may include reaction conditions, photocatalyst activity, bandgap energy, light source and process parameters, and the comparison between different methods is shown in Table 2 [31-38]. It is clear that the Cu2O/TiO2 heterojunctions prepared in this work have better CH3OH yields under visible light irradiation.”

 (The related statements are also described in the revised manuscript)

Comment #4: what was the size of TiO2 nanoparticles? Can authors provide the morphological image of TiO2 nanoparticles?

L158: what is meant by "pure TiO2"? Is it nanoparticles.?

L176: Average size or size range?"

Response:

L158 (L182 (revised)):  The “pristine” TiO2 nanoparticles are described in the revised manuscript. The related statements were revised: “After a 6-h UV-vis irradiation, 70.6, 61.8, 21.0 and 12.4 µmol/g-catalyst methanol are formed by the Cu2O/TiO2 heterojunctions having the XCu2O fractions of 0.1, 0.2, 0.5 and pristine TiO2 nanoparticles, respectively."

L176 (L219 (revised)): ”The Cu2O (in Cu2O/TiO2 heterojunction) and TiO2 have nanoparticle diameters of ~5 and 70-130 nm, respectively.”

The HRTEM image of the sample in Figure 2(d) shows lattice fringes spacing of 0.212 and 0.237 nm, corresponding to the (200) and (111) plane of Cu2O, respectively [28]. The TiO2 with a highly crystallinity has the d-spacings of 0.352 and 0.246 nm related to the (100) and (004) planes of anatase TiO2 [28].

(The related statements are also described in the revised manuscript)

Figure. 3 The TEM images of the (a) pristine TiO2 and (b) Cu2O/TiO2 heterojunctions (XCu2O=0.5) with (c) EDS mapping and (d) HRTEM image.

Comment #5: L88: How did authors conclude "In a solid-liquid reaction system involving dissolved CO2 in water, ·CO2-, ·COOH, HCOOH and HCHO are the intermediate species? -Please justify this statement.  

Response:

Intermediate species such as ·CO2-, ·COOH, HCOOH and HCHO were frequently found in photocatalytic reduction of dissolved CO2 in water [18]. CO2 may be activated by a one-electron transfer step and form the ·CO2- radical ion. The ·CO2- may be reduced to yield the hydroxyformyl radical (·COOH), which recombines a hydrogen radical (H+) and an electron (e-) to form formic acid [18]. In the following step, formic acid accepts H+ and e- to form formaldehyde. Formic acid and formaldehyde seem to be formed prior to the methanol generation. Thus, the key points that control the photocatalytic CO2-to-CH3OH may include reaction conditions, photocatalyst activity, bandgap energy, light source and process parameters, and the comparison between different methods is shown in Table 2 [31-38]. It is clear that the Cu2O/TiO2 heterojunctions prepared in this work have better CH3OH yields under visible light irradiation. We found the intermediate species during photocatalytic reduction of CO2 by in situ FTIR in separated experiments.

(The related statements are also described in the revised manuscript)

Comment #6: Analysis of surface area in Photocatalysis is important. Did authors think of determining surface area of the catalyst? N2-BET analysis is a good technique to perform such test. Are these synthesized Cu2O-TiO2 heterojunctions porous or non-porous? Include this information.

Response:

Additional physicochemical characterization (BET and zpc) of the photocatalysts was studied. The nitrogen adsorption-desorption isothermals of the pristine TiO2 and Cu2O/TiO2 heterojunctions are shown in Figure 4(A). The absorption isothermals of the Cu2O/TiO2 heterojunctions can be classified as type IV with H1 hysteresis loops, suggesting that they have mesoporous structure. In Figure 4(B), the Cu2O/TiO2 heterojunctions have greater pore volumes with the pore diameters between 10-40 nm than the pristine TiO2 possessing relatively high pore diameters of 30-70 nm. It seems that the smaller Cu2O nanoparticles may, to some extent, be incorporated in the pores of the TiO2, which creates more internal surfaces in the interfaces of Cu2O and TiO2 nanoparticles. In Table 1, as expected, the Cu2O/TiO2 heterojunctions have relatively high specific surface areas (94-120 m2/g) and small average pore diameter, which may benefit the photocatalytic reduction of CO2 to methanol.

 (The related statements are also described in the revised manuscript)

Figure 4. (A) N2 absorption-desorption isothermals and (B) pore size distributions of the Cu2O/TiO2 heterojunctions having the XCu2O fractions of (a) 0.1, (b) 0.2, (c) 0.5 and (d) pristine TiO2.

Table 1. The BET surface areas, average pore diameters, and zeta potential of the pristine TiO2 and Cu2O/TiO2 heterojunctions having the XCu2O fractions of 0.1-0.5.

Photocatalysts

XCu2O

fractions

BET surface

Area (m2/g)

BJH average pore

diameter (nm)

Zeta potential (mV)

Pristine TiO2

0

48

50

52.7

Cu2O/TiO2

0.1

94

23

-21.1

Cu2O/TiO2

0.2

105

15

-26.6

Cu2O/TiO2

0.5

120

15

-39.1

Comment #7: Can authors include some information about the morphology of the TiO2 samples in the main manuscript. Provide the size distribution and images of the materials.) 

Response:

The TEM images of the TiO2 and Cu2O/TiO2 heterojunctions are shown in Figure. 3. It is clear that the Cu2O (in the Cu2O/TiO2 heterojunction) and TiO2 have nanoparticle diameters of ~5 and 70-130 nm, respectively.

(The related statements are also described in the revised manuscript)

Figure. 3 The TEM images of the (a) pristine TiO2 and (b) Cu2O/TiO2 heterojunctions (XCu2O=0.5) with (c) EDS mapping and (d) HRTEM image.

Comment #8: Authors need to cite relevant article for the discussion e.g., –

- XRD:Anatase crystal of TiO2 and JCPDS card#; and Cu2O crystal information & JCPDS#

- Scherrer’s equation

- UV-Vis:Tauc plot & Kubelka-Munk equation?

Response:

The diffraction peaks at 29.6°, 36.5°, 42.4°, 61.4°, 73.6° and 77.5° correspond to (110), (111), (200), (220), (311) and (222) phases of the crystalline Cu2O (JCPDS card No. 78-2076), respectively [25]. A high intensity diffraction peak at 36.4° confirms the existence of Cu2O in the Cu2O/TiO2 heterojunctions. Other diffraction peaks at 25.3°, 37.8°, 48.0°, 53.9°, 55.0°, 62.7°, 68.8°, 70.3° and 75.0° can be indexed to (101), (004), (200), (105), (211), (204), (116), (220) and (215) planes of TiO2 (JCPDS card No. 71-1167) associated with the anatase phase [26], indicating that Cu2O/TiO2 heterojunctions consist of anatase predominantly.

The Scherrer (t=kλ/Bcosθ) and Kubelka-Munk (αhν=A(hν-Eg)n) equations are expressed in the Experimental section.

(The related statements are also described in the revised manuscript)

Comment #9: Also. Fig 2(d) is not clear. Authors should provide a better HRTEM image to showcase the heterojunctions of Cu2O-TiO2. Authors need to provide the EDX mapping of the Cu2O-TiO2 heterojunctions?

Response:

The scale-bar visibility in Figure 3(a-d) was improved. The HRTEM image of the sample in Figure 2(d) shows lattice fringes spacing of 0.212 and 0.237 nm, corresponding to the (200) and (111) plane of Cu2O, respectively [28]. The TiO2 with a highly crystallinity has the d-spacings of 0.352 and 0.246 nm related to the (100) and (004) planes of anatase TiO2 [28].

(The related statements are also described in the revised manuscript)

Figure. 3 The TEM images of the (a) pristine TiO2 and (b) Cu2O/TiO2 heterojunctions (XCu2O=0.5) with (c) EDS mapping and (d) HRTEM image.

Comment #10: What was the reference sample used in this study? Can authors compare and highlight their results on methanol oxidation via photocatalysis with the relevant literature. That would give a better perspective of the importance of this work. Can authors include a tabular form or a brief highlight in the results and discussion part?

Response:

Intermediate species such as ·CO2-, ·COOH, HCOOH and HCHO were frequently found in photocatalytic reduction of dissolved CO2 in water [18]. CO2 may be activated by a one-electron transfer step and form the ·CO2- radical ion. The ·CO2- may be reduced to yield the hydroxyformyl radical (·COOH), which recombines a hydrogen radical (H+) and an electron (e-) to form formic acid [18]. In the following step, formic acid accepts H+ and e- to form formaldehyde. Formic acid and formaldehyde seem to be formed prior to the methanol generation. Thus, the key points that control the photocatalytic CO2-to-CH3OH may include reaction conditions, photocatalyst activity, bandgap energy, light source and process parameters, and the comparison between different methods is shown in Table 2 [31-38]. It is clear that the Cu2O/TiO2 heterojunctions prepared in this work have better CH3OH yields under visible light irradiation.

 (The related statements are also described in the revised manuscript)

Table 2. Method comparison for photocatalytic CO2-to-CH3OH.

Photocatalyst

Light source

Bandgap

Energy (eV)

Reactions

CH3OH yield

(µmol/g/h)

Ref.

Cu2O/TiO2

UV

Vis

2.9-3.0

100 mg photocatalysts,

CO2 in deionized water

9-13

12-70

This work

Co/TiO2

UV

-

100 mg photocatalysts,

CO2 in NaHCO3 (1 M)

0.05

[31]

Anatase TiO2

Vis

2.9-3.2

500 mg photocatalysts,

CO2 in deionized water

2.74

[32]

SnO2/g-C3N4

UV

-

20 mg photocatalysts,

CO2 in water vapor

0.02

[33]

rGO/ZnO

Vis

2.8

100 mg photocatalysts,

CO2 in water vapor

0.42

[34]

rGO/Cu2O

UV

2.7-2.8

100 mg photocatalysts,

CO2 in NaOH (1 M)

8.77

[35]

CQDs/Cu2O

Vis

2.4-2.6

150 mg photocatalysts,

CO2 in deionized water

1.96

[36]

ZnTe/SrTiO3

UV

2.8-3.4

20 mg photocatalysts,

CO2 in deionized water

0.75

[37]

ZnO/g-C3N4

UV

2.6-3.0

10 mg photocatalysts,

CO2 in deionized water

0.06

[38]

Reviewer 3 Report

Chen et al. demonstrated their finding on “Photocatalytic reduction of CO2 to methanol by Cu2O/TiO2 heterojunctions”. In the literature, there are many research articles about Cu2O/TiO2 hybrid structures for photocatalytic applications. The submitted manuscript might be beneficial to understand the effect of the illuminated light (UV-Vis or Vis) on the photocatalytic performance. However, it has many drawbacks especially presenting the data and supporting them with previous literature knowledge.  However, it might be dense revision after addressing of following suggestions.

  1. “A variety of strategies, such a metal ion doping…….. to the visible light region” The cited articles are pretty old. Please cite some recent articles about heterostructures.
  • ACS Omega2016, 1, 5, 868–875
  • ACS Omega2019, 4, 2, 3392–3397
  • Nanoscale, 2019,11, 9840-9844
  • Sci. Technol., 2016,6, 7967-7975
  1. Authors should do detailed literature search on “why Cu2O should be used” and “how to use Cu2O”.
  2. Schematic would be beneficial to better understand the synthesis steps.
  3. Figure 2 should be changed with better resolution. It is hard to see and distinguish the nanoparticles.
  4. There is not any literature support in result part (XRD, TEM, UV-vis spectra, Tauc plot, photoluminance, and photocatalytic reduction of CO2 to methanol). All results (especially proposed mechanism) should be supported by literature.

Author Response

Review of Manuscript ID: Sustainability- 1474103

Title: Photocatalytic reduction of CO2 to methanol by Cu2O/TiO2 hetero-junctions

Responses to the Comments of Reviewers

The manuscript was carefully re-examined and revised to responses and fulfilled the reviewers’ comments. Additional data, better presentation of Figures, and more detailed discussion are included in the revised manuscript. All the authors have seen the revised manuscript with the responses to reviewers’ comments and approved submission. The manuscript was presented at the International Conference on Innovations in Energy Engineering & Cleaner Production (IEECP’21) and caught notable attention. Thus, we believe this manuscript will be of interest to the Sustainability audience.

Reviewer #3

Comment #1: “A variety of strategies, such a metal ion doping…….. to the visible light region” The cited articles are pretty old. Please cite some recent articles about heterostructures.

ACS Omega2016, 1, 5, 868–875.

ACS Omega2019, 4, 2, 3392–3397

Nanoscale, 2019,11, 9840-9844

Sci. Technol., 2016,6, 7967-7975

Response:

The cited articles related to heterostructures and promotion of photocatalysts were updated, which was expressed in the revised manuscript.

Comment #2: Authors should do detailed literature search on “why Cu2O should be used” and “how to use Cu2O”.

Response:

To effectively suppress the rapid recombination of photoexcited electron and hole, the heterojunction structure could facilitate the electron migration [18, 19]. Cuprous oxide (Cu2O), a typical p-type semiconductor, has wide applications in solar cells, photocatalysis, and Hydrogen evolution reaction (HER) [21]. Cu2O having bandgap energy of 2.0-2.2 eV could effectively harvest visible light for photocatalysis. Although, the photocatalytic CO2-to-CH3OH by Cu2O is thermodynamically feasible, its CH3OH yield suffered from the low solar-conversion efficiency [22]. By heterojunction between p-type Cu2O and n-type TiO2, the recombination of photo-excited charges could be effectively retarded and facilitated photocatalytic reactions [23, 24]. In this work, the novel Cu2O/TiO2 heterojunctions were thus prepared by the simple soft chemical method as the visible light photocatalysts to extend absorption to the visible light range for enhanced photocatalytic reduction of CO2 to methanol.

(The related statements are also described in the revised manuscript)

Comment #3: Schematic would be beneficial to better understand the synthesis steps.

Figure 2 should be changed with better resolution. It is hard to see and distinguish the nanoparticles.

Response:

As shown in Figure 1, Cu2O was prepared by the facile soft chemical method. Briefly, CuCl2 (97%, Merck) (10.1 mmol) was dispersed in a NaCl solution (5 M) (100 mL) with a dispersant (polyethylene glycol 20,000 (Alfa Aesar) (0.025 mmol)), which was stirred at 298 K for 1 h. Na3PO4 (96%, Aldrich) (9.76 mmol) was added to the solution and stirred for 1 h. Cu2O was centrifuged and cleaned with distilled water and ethanol for three times. Titanium butoxide (Ti(OBu)4) (97%, Aldrich) and Cu2O at the XCu2O mole fractions (Cu2O/(Cu2O + TiO2)) of 0.1, 0.2 and 0.5 were mixed in deionized water, which were centrifuged, dried at 378 K for 4 h and heated at 723 K under a N2 flow (99.99%) (20 mL/min) for 2 h to obtain the Cu2O/TiO2 heterojunctions for photocatalysis experiments.

Figure. 1 Preparation procedure for the Cu2O/TiO2 heterojunction photocatalysts.

The TEM images of the TiO2 and Cu2O/TiO2 heterojunctions are shown in Figure 3. It is clear that the Cu2O (in the Cu2O/TiO2 heterojunction) and TiO2 have nanoparticle diameters of ~5 and 70-130 nm, respectively. The presence of Cu, Ti, and O in the Cu2O/TiO2 heterojunction could be revealed by energy-dispersive x-ray (EDX) spectroscopy (see Figure 2(c)). The HRTEM image of the sample in Figure 3(d) shows lattice fringes spacing of 0.212 and 0.237 nm, corresponding to the (200) and (111) plane of Cu2O, respectively [22]. The TiO2 with a highly crystallinity has the d-spacings of 0.352 and 0.246 nm related to the (100) and (004) planes of anatase TiO2 [24].

Figure. 3 The TEM images of the (a) pristine TiO2 and (b) Cu2O/TiO2 heterojunctions (XCu2O=0.5) with (c) EDS mapping and (d) HRTEM image.

Comment #4: There is not any literature support in result part (XRD, TEM, UV-vis spectra, Tauc plot, photoluminance, and photocatalytic reduction of CO2 to methanol). All results (especially proposed mechanism) should be supported by literature.

Response:

More updated references related to the results from XRD, TEM, UV-vis spectra, Tauc plot, photoluminance, and photocatalytic reduction of CO2 to methanol were included in the revised manuscript.

XRD: “The diffraction peaks at 29.6°, 36.5°, 42.4°, 61.4°, 73.6° and 77.5° correspond to (110), (111), (200), (220), (311), and (222) phases of the crystalline Cu2O (JCPDS card No. 78-2076), respectively [25]. A high intensity diffraction peak at 36.4° confirms the existence of Cu2O in the Cu2O/TiO2 heterojunctions. Other diffraction peaks at 25.3°, 37.8°, 48.0°, 53.9°, 55.0°, 62.7°, 68.8°, 70.3° and 75.0° are indexed to (101), (004), (200), (105), (211), (204), (116), (220), and (215) planes of TiO2 (JCPDS card No. 71-1167) having the anatase phase [26], indicating that Cu2O/TiO2 heterojunctions consist of anatase predominantly.”

TEM: “The TEM images of the TiO2 and Cu2O/TiO2 heterojunctions are shown in Figure 3. It is clear that the Cu2O (in the Cu2O/TiO2 heterojunction) and TiO2 have nanoparticle diameters of ~5 and 70-130 nm, respectively. The presence of Cu, Ti, and O in the Cu2O/TiO2 heterojunction could be revealed by energy-dispersive x-ray (EDX) spectroscopy (see Figure 3(c)). The HRTEM image of the sample in Figure 3(d) shows the lattice fringes spacing of 0.212 and 0.237 nm, corresponding to the (200) and (111) plane of Cu2O, respectively [28]. The TiO2 with a highly crystallinity has the d-spacing of 0.352, 0.246 nm, related to the (100), (004) plane of anatase TiO2 [28].”

Figure. 3 The TEM images of the (a) pristine TiO2 and (b) Cu2O/TiO2 heterojunctions (XCu2O=0.5) with (c) EDS mapping and (d) HRTEM image.

UV-vis spectra: “The diffuse reflectance ultraviolet-visible spectra of the Cu2O/TiO2 photocatalysts were determined. In Figure 5, the absorbance of the Cu2O/TiO2 heterojunctions in the range of 200-800 nm is observed. Compared with TiO2, the fundamental absorbance cuts at 400 nm, and TiO2 mixed with Cu2O revealed a significantly enhanced absorption in the visible light region. It is clear that TiO2 with Cu2O causes a red-shift to 400-800 nm, in the visible light range, possibly due to the fact of forming the Cu2O/TiO2 heterojunctions [28].“

Tauc plot: “In Figure 6, the bandgap energies of the photocatalysts were determined by the Kubelka-Munk transforms [29]. The direct band-gaps of the TiO2, Cu2O and Cu2O/TiO2 heterojunctions having the XCu2O fractions of 0.1, 0.2 and 0.5 were estimated to be 3.20, 2.09, 3.03, 3.0 and 2.94 eV, respectively. It seems that the coupling of Cu2O with TiO2 can effectively decrease their bandgap energies. The Cu2O/TiO2 heterojunctions turn out to be more active than TiO2 under visible light illumination”.

Photoluminance: “Charge separation efficiency of photoinduced electrons and holes is also one of important factors in photocatalysis. The photoluminescence spectra can provide the information of charge carrier trapping, migration and transfer [30]. The photoluminescence spectra of Cu2O/TiO2 heterojunctions are shown in Figure 7. The PL intensity of Cu2O/TiO2 heterojunctions is less than that of TiO2. A clear quenching of the photoluminescence emission of the Cu2O/TiO2 heterojunctions is observed, especially for the Cu2O/TiO2 heterojunctions having the XCu2O fractions of 0.2, a maximum quenching. Such a quenching of the photoluminescence suggests that the separation of photogenerated electron and hole pairs in the Cu2O/TiO2 heterojunctions has been efficiently improved. Cu2O can transfer the photo-generated holes from TiO2 to inhibit the recombination of photo-generated electrons and holes significantly, and may consequently lead to an enhanced photocatalytic activity.

Photocatalytic reduction of CO2 to methanol: Figure 8(A) shows the yield of CH3OH from photocatalytic reduction of CO2 by the Cu2O/TiO2 heterojunctions under visible light irradiation. After a 6-h visible light irradiation, the yields of CH3OH photocatalyzed by the Cu2O/TiO2 heterojunctions with the XCu2O fractions of 0.1, 0.2, and 0.5 are 9.29, 11.44, and 13.06 µmol/g-catalyst, respectively. Note that as expected, TiO2 is not very effective for the photocatalytic CO2-to-CH3OH. Also, more visible light active Cu2O in the Cu2O/TiO2 heterojunctions favors the formation of methanol. On the other hand, under UV-vis irradiation for 6 h, CH3OH yields from the photocatalytic CO2-to-CH3OH by the Cu2O/TiO2 heterojunctions are 21.0-70.6 µmol/g-catalyst (see Figure 8(B)). It is clear that the yields of CH3OH under UV-vis irradiation is greater than that under visible irradiation. In addition, the yield of CH3OH decreases with an increase of the Cu2O fraction in the Cu2O/TiO2 heterojunctions. It seems that excess Cu2O in Cu2O/TiO2 heterojunctions may lead to less UV-light exposure of photocatalysts, and may decrease the conversion efficiency of CO2 to CH3OH.

Figure. 8 Photocatalytic reduction of CO2 to methanol under (A) visible and (B) UV-vis irradiation by the Cu2O/TiO2 heterojunctions having the XCu2O fractions of (a) 0.1, (b) 0.2, (c) 0.5, and (d) TiO2 nanoparticles.

Round 2

Reviewer 1 Report

The authors revised their work according to the suggestions

Reviewer 3 Report

The authors made significant changes in the previous version. The revised version should be accepted for publication.